# A Review of Chronic Pain and Cognitive, Mood, and Motor Dysfunction Following Mild Traumatic Brain Injury: Complex, Comorbid, and/or Overlapping Conditions?

**DOI:** 10.3390/brainsci7120160

**Published:** 2017-12-06

**Authors:** Ramesh Grandhi, Samon Tavakoli, Catherine Ortega, Maureen J. Simmonds

**Affiliations:** 1Department of Neurosurgery, University of Texas Health San Antonio, 7703 Floyd Curl Drive, MC 7843, San Antonio, TX 78229-3900, USA; tavakolis@uthscsa.edu; 2Department of Physical Therapy, University of Texas Health San Antonio, San Antonio, TX 78229, USA; ortegac2@uthscsa.edu (C.O.); simmondsm@uthscsa.edu (M.J.S.)

**Keywords:** traumatic brain injury, concussion, pain, cognition, mood, movement

## Abstract

Mild traumatic brain injury (mTBI) is commonly encountered in clinical practice. While the cognitive ramifications of mTBI are frequently described in the literature, the impact of mTBI on emotional, sensory, and motor function is not as commonly discussed. Chronic pain is a phenomenon more prevalent among patients with mTBI compared to those with moderate or severe traumatic brain injury. Chronic pain can become a primary disorder of the central nervous system (CNS) expressed as widespread pain, and cognitive, mood, and movement dysfunction. Shared mechanisms across chronic pain conditions can account for how pain is generated and maintained in the CNS, irrespective of the underlying structural pathology. Herein, we review the impact of mTBI on cognitive, emotional, sensory, and motor domains, and the role of pain as an important confounding variable in patient recovery and dysfunction following mTBI.

## 1. Introduction

Mild traumatic brain injury (mTBI) is the most common neurological injury; it has an estimated incidence of 600/100,000 according to the World Health Organization Collaborating Center Task Force on mTBI [1]. While mTBI is commonly encountered in clinical practice, its definition remains as varied as its clinical presentation. Classically, the injury has been defined by initial patient presentation with a Glasgow Coma Scale (GCS) score greater than or equal to 13, loss of consciousness lasting less than 30 min, and post-traumatic amnesia lasting less than 24 h. Symptoms such as headache, nausea, vomiting, dizziness, imbalance, fatigue, blurred vision, memory difficulty, and concentration difficulty can resolve within days; however, it is not uncommon for symptoms to last for months post-injury [2]. A proportion of patients in the mTBI population continue to experience an array of symptoms, collectively referred to as post-concussion syndrome (PCS), that may persist for more than a year after initial insult [3].

Although the initial impact of mTBI may be minimal and transient, emerging evidence from the fields of sports, aging, and military medicine indicate that mTBI can be associated with a multiplicity of persistent problems that may be subtle or substantive. Hence, mTBI can result in a lifetime impact on cognitive, emotional, sensory, and motor domains [4,5], as well as mimic or accelerate age-related physiological, physical, psychosocial, and cognitive changes. 

Chronic pain is particularly problematic in patients following mTBI; it can occur independently of concomitant diagnoses of depression and post-traumatic stress disorder (PTSD) [6,7]. These findings have given credence to the theory that chronic pain can become a primary disorder of the central nervous system (CNS) when it is expressed as widespread pain and cognitive, mood, and movement dysfunction [8,9]. Previous work detailing the association between chronic pain and persons with TBI has demonstrated a prevalence of 51.5% in civilian patients compared to 43.1% amongst veterans, with a significantly higher rate of chronic pain in patients with mTBI (75.3%) compared to those with moderate or severe TBI (32.1%) [7]. It appears that, separately and together, mTBI and chronic pain are associated with complex symptom clusters and the disturbance of cognitive, physical, and psychological function. Moreover, due to pain and mTBI, these clinical presentations may have overlapping and interactive mechanisms and potentially exponential effects on symptom clusters, mood, and movement. The purpose of this manuscript is to review the literature published on mTBI and chronic pain in order to evaluate their comorbid effects. By better understanding the implications of pain in persons with mTBI along with the incidence, aggravation, and management of complex symptoms and comorbidities, we can enhance patient care, reduce morbidity, and improve long-term outcomes for this particular population.

## 2. Consequences of Mild TBI

As previously noted, patients may experience significant problems following mTBI spanning cognitive, emotional, and motor domains. Symptoms of mTBI include fatigue, headaches, visual and vestibular disturbances, memory loss, poor attention/concentration, sleep disturbances, mood disorders, and seizures. Previously, a small percentage (10–25%) of patients suffering from mTBI were believed to experience persistent PCS at three months or more post-injury [10,11]. However, other studies have demonstrated higher rates of persistent PCS [12,13]. 

In a prospective analysis looking for early predictors for PCS in mTBI patients, physical symptoms were the most prevalent early on. Emotional and cognitive symptoms were less prevalent early on, but they were relatively more persistent at three months post-injury. The delayed expression of emotional and cognitive symptoms can be attributed to the impact of physical symptoms on function, and are alleviated by activity and improvement in function [14]. The strongest predictors of PCS at three months were anxiety, noise sensitivity, and difficulty thinking. PCS is commonly associated with female patients, preinjury depression, early depression, memory problems, irritability, and light sensitivity; many of the common acute signs and symptoms, such as headache, nausea, vomiting, dizziness, and fatigue, provided poor predictive power [12,15]. Further study of predictive factors for PCS could potentially help target individuals for early intervention.

A myriad of cognitive effects occur as a result of mTBI, including deficits in executive functioning, learning, memory, attention, and processing speed. Many studies have demonstrated an acute decrease in performance on a variety of cognitive processing tests in patients with mTBI up to one week post-injury when compared to non-mTBI controls [11,16,17,18,19]. More recently, Landre et al. (2006) found that mTBI subjects performed worse on all their tested cognitive measures when compared to non-mTBI trauma patients [6]. In a recent review including 45 studies on mTBI and chronic mild cognitive impairment reported more than three months post-injury, McInnes et al. (2017) estimated that 58% of adults with mTBI exhibited persistent cognitive impairment [13]. Age appears to influence the nature and magnitude of cognitive sequelae such that infants and preschoolers with mTBI exhibit linguistic changes affecting expressive language and Verbal IQ [20]. It has similarly been reported that children with mTBI have impaired executive functioning and attention one year post-injury when compared to non-injured children [21]. Some studies also report worse cognitive outcomes for children one year post-injury when compared to adults [22].

Individuals with mTBI are at increased risk for persistent mood disturbances. Emotional consequences contribute to the clinical challenge of managing patients with mTBI; they often manifest as increased irritability, depression, anxiety, and sleep disturbances [12]. Mood and affect changes can manifest acutely or develop over time. Jorge et al. (2006) hypothesized that neuronal and glial loss within the hippocampus might progress after the initial injury. Investigators found that patients who developed mood disorders had lower hippocampal volumes than patients without mood disorders [23]. They also reported significant interaction between the diagnosis of a mood disorder and the severity of TBI. Multiple studies have further elucidated the prevalence of mood disorders in patients with mTBI. In one meta-analysis reviewing nine studies with a mean aggregate follow-up of roughly 12 months, depression was the most frequently described emotional symptom (52.9%), followed by anxiety (29.4%) [24]. Sleep alterations are also present in this population, which may further exacerbate emotional symptoms. Patients with mTBI have less rapid eye movement (REM), longer REM latency, and more sleep complaints [25]. Taken in context with sleep quality and quantity having a significant influence on the development and persistence of pain, mood, and movement disorders, these findings not only highlight an important consequence of mTBI but also suggest that disorders of sleep, pain, mood, and movement may have shared mechanisms and exponential effects in patients with mTBI. 

Movement and postural control have been assessed in individuals with moderate TBI; however, to date, less attention has been paid to those with mild TBI. Interestingly, individuals with moderate TBI exhibit movement patterns that are temporally and spatially similar to those of individuals with spinal dysfunction and stroke [26,27] across a range of tasks. Clinically, these patterns manifest as relatively slow, self-selected walking speeds, lower push-off forces, and greater lateral excursion of the center of pressure. These changes may be attributable to reduced agonist forces and a concomitant increase in antagonist forces during movement. Though these changes in motor control and motor performance may be present, their impact on performance may not be detected during standard clinical assessments. Preliminary data from our lab suggests that dual task (simultaneous physical and cognitive performance) testing shows greater promise as an indicator of performance compromise due to mTBI than single task performance alone (unpublished data). Persistent motor problems exhibited as slowed motor execution speed and compromised postural control have been demonstrated following sports-related concussion in football players nine months post-concussion [5]. Moreover, it now appears that the detrimental effects of sports concussions on motor function can persist and/or become evident decades later, partly due to diminished reserve capacity and a decline in physiological, physical, and cognitive systems [28].

## 3. Chronic Pain in Mild TBI

Chronic pain is particularly problematic and relatively neglected in persons with mTBI [29,30]. Although pain is thought to be more common in those who sustained their brain injury from violent trauma, as previously mentioned, it is actually more prevalent following mTBI than in cases involving moderate or severe TBI. The most commonly encountered manifestation of pain following TBI is headache, with a prevalence of 57.8% [7]. However, recent studies have identified other pain problems, such as neck pain, back pain, and musculoskeletal pain initially generated from various body parts [31].

Regardless of its genesis, it is now recognized that chronic pain can become a primary disorder of the CNS expressed as widespread pain and cognitive, mood, and movement dysfunction [8,9]. Shared mechanisms across chronic pain conditions can account for how pain is generated and maintained in the CNS, irrespective of the underlying structural pathology [8,32]. The hallmark of ‘centrally driven’ pain conditions is a diffuse hyperalgesic state (identifiable via experimental sensory testing), multifocal pain, fatigue, insomnia, memory difficulty, and a higher rate of comorbid mood disorders. 

### 3.1. Neurobiology of Pain in mTBI

Until recently, central pain following TBI, especially mTBI, had not been studied in depth. In 2007, Ofek and Defrin tested 15 patients with TBI who were suffering from chronic central pain and compared them to 16 pain-free persons with TBI and a healthy, pain-free control group [29]. The authors reported that chronic pain developed at a relatively late onset (6.6 ± 9 months). They described it as almost exclusively unilateral and characterized it as pricking, throbbing, and burning. They also found that both TBI groups had reduced thermal and tactile sensations and very high rates of allodynia, hyperpathia, and exaggerated wind-up compared to controls, particularly in the painful regions. Such characteristics resemble those of other chronic pain conditions that have central pain-sensitizing components, such as fibromyalgia [33,34]. Recent evidence has underscored the influence of altered or dysregulated endogenous neurotransmitter systems in fibromyalgia [35]. Similarly, upregulation of inflammatory pronociceptive factors has been identified in neuropathic pain [36]. In turn, differential expression in neurotransmitters and modulators results in changes to pain processing, which leads to pain facilitation, nociception, and attenuation in endogenous pain inhibitory pathways. Animal models of mTBI have similarly demonstrated that changes in the expression of similar neurotransmitters have resulted in central pain sensitization [37]. This suggests that neuronal hyperexcitability or augmented central pain processing may also be a contributing factor to chronic pain in mTBI. Interestingly, Tan et al. (2009) found dysregulation of the autonomic nervous system demonstrated by depressed heart rate variability among veterans with concomitant diagnoses of mTBI, PTSD, and chronic pain, which suggests that there are additional neurophysiologic consequences of mTBI and pain [38]. The possible synergistic effects of these diagnoses merit further analysis.

### 3.2. Pain-Related Cognitive, Mood, and Movement Dysfunction in mTBI

The effects of pain on cognitive functions, such as attention and memory, have already been established [39,40]. For example, pain can demand cognitive attention, thereby reducing available cognitive reserves for other functions. Dick and Rashiq (2007) reported that individuals with chronic pain who were challenged with a neuropsychological test of attention demonstrated impaired attention independent of age or education level [41]. Other studies have shown that pain may impair performance on cognitively challenging tasks [6,42]. However, the majority of studies show no difference in performance when patients with mTBI are compared to non-brain injured controls with similar levels of pain [43,44,45].

Pain is multidimensional and includes sensory, affective, and cognitive components. Chronic pain is associated with psychopathology; moreover, psychopathologies are recognized as being a consequence of inadequate and inappropriate management of persistent pain and its impact [46,47]. However, it is recognized that chronic pain may modulate and even potentiate cognitive and mood dysfunction in patients who have experienced a concussion and vice versa [48]. Jamora and colleagues (2013) found that intensity of pain correlated with the degree of emotional disturbance a patient experienced, which suggests a strong association between pain and mood symptoms [44]. Although chronic pain is a common complication of TBI and may be independent of psychological disorders such as depression and PTSD in adults [7], pre-existing depression, PTSD, or sleep disturbances also play a crucial role in the development and negative impact of chronic pain [11]. Amongst adolescents, female sex and more depressive symptoms at three months post-injury were both found to have a positive correlation with persistent pain [49].

Headache in the acute post-TBI period impacts mood and sleep quality [50]. Amongst patients with mTBI, Suzuki et al. (2017) showed that patients with moderate-to-severe pain required more sleep and naps compared to those with mild-to-no pain at one month post-injury [51], which suggests either a need for increased sleep to alleviate pain or a decreased quality of sleep in patients with mTBI. 

In individuals with chronic pain, illness, and injury, our group has previously shown that, regardless of disease or disability, individuals tend to exhibit a generalized psychomotor slowing and a ‘stiffening’ during movement that is akin to aging-related movement patterns [52,53,54]. In a sample of veterans with chronic pain, we recently documented average aging effects on physical and cognitive function of between 30–40 years and 10–15 years, respectively [55]. Noteworthy is the fact that multi-modal (e.g., education, activity, social support) non-pharmacological interventions can improve cognitive and physical function as well as mood [56].

## 4. Conclusions

Overall, there is an abundance of evidence that shows that both mTBI and chronic pain are complex problems that impact mood, cognitive function, and physical function. It is also clear that both problems have shared neurobiological CNS mechanisms that underlie and account for their complexity in symptom and functional impact. To date, it is not clear to what extent the problems encountered are comorbid rather than overlapping, related conditions. Regardless, the influence of mTBI and pain on cognition, mood, and movement points to a complex interaction of factors. An understanding of the relationships and interactions between and among components of injury, pain, mood, cognition, movement, and their directional pathways is currently incomplete. This is not surprising, given the complexity of the underlying neuromatrix and the integration of the involved sensory, motor, and emotional systems that will have a differential impact on the mind, mood, and body of those with mTBI. An expanded conceptualization of the problems engendered in patients following mTBI and their impact is important. Furthermore, understanding the personal, physiological, psychological, and sociological factors that influence the expression of potentially subtle mTBI symptoms, as well as the patient’s response to those symptoms and the potential for progression to chronic problems of distress and disability, is essential to meaningful, efficient, and effective interventions.

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
