# Peer review of "A Review of Chronic Pain and Cognitive, Mood, and Motor Dysfunction Following Mild Traumatic Brain Injury: Complex, Comorbid, and/or Overlapping Conditions?"

_brainsci, 2017, doi:10.3390/brainsci7120160_

Round 1

Reviewer 1 Report

Many of the references cited in this paper are more than 5 years old - and in a subject such a mTBI which has been written about extensively in recent years - this suggests that all of the information given in this paper may not be up to date. I realise that this is a subject that has been studied intensively by one of the authors, but although the underlying message does come through I do not fel that the paper flows well. What in feact is the overall message you wish to relate from this paper and what are the clinical implications that can be drawn.

Author Response

Dear Reviewer,

We sincerely appreciate the opportunity to strengthen our paper based on your suggestions. First, we have amended the title of our manuscript to deliver the message that chronic pain and dysfunction following mTBI in cognitive, mood, and motor domains may very well be comorbid or overlapping conditions. Second, we further highlighted this by expanding the introduction and including a sentence in which we underscore that the purpose of our review was to delve into this complex relationship further. In addition, we have added narrative and references to strengthen the fact that these conditions can exist independently but also have shared explanatory neurobiological mechanisms that account for the complex clinical problems encountered.

Finally, with regards to the "current-ness" of the manuscript, we have added more references and have also touched on some of the work that we are presently doing in our lab with reference to an ongoing clinical study with our mild TBI patients and our preliminary findings.

Reviewer 2 Report

This is an important paper that deserves consideration. Pain is understudied among TBI, especially its relationship with cognition, mood, and motor function. Please consider the following:

1. The title is misleading--does this paper really examine the influence of pain on cognition, mood, and motor function? It is a review and I feel the title makes it seem like a study. Perhaps make it clear in the title that this is a brief review of the literature

2. This is not a meta-analysis nor a systematic review. I think the authors should make it clear the purpose of this paper. I think there should be a purpose statement at the end of the introduction section highlighting the objective of this review. 

3. In the conclusion section, please add more regarding the potential for future research. What can be learned from this brief review? What should future focus on? What kind of studies are required? I think this is very important. You want the readers to understand where current research stands and what needs to come next.

4. First time 'central nervous system' is mentioned in the paper, an acronym is not added. The CNS acronym is added at the second mention

Author Response

Dear Reviewer,

We appreciate the opportunity to strengthen our manuscript based on your suggestions. We trust that our revised manuscript addresses your concerns.

With regards to suggestion #1, we have changed the title of our paper to better reflect the content and the fact that this is a review paper.

To address points #2 and #3 that you cited in your review of our initial manuscript, we have added a purpose statement at the end of the introduction section to underscore the objective of our review. We also expanded our conclusion section to re-summarize and highlight what is presently understood and the fact that there are many unknowns with regards to pain and patients with cognitive, mood, and motor dysfunction following mTBI.

We have edited our manuscript to address point #4.

Reviewer 3 Report

The authors provide a well written review on cognitive, mood, motor function and pain sequelae after mild traumatic brain injury. Given the title, I expected to see more data and narrative regarding the associations between post-traumatic pain and pertubations of cognition, mood, and movement. These associations are presented somewhat superficially, without data. Whether treatment of pain improves these changes was not addressed. These omissions represent the major weaknesses of the paper; addressing them would result in a much higher quality paper with increased interest and "bang for the buck" in terms of reader's time.

Author Response

Dear Reviewer, 

We appreciate the opportunity to strengthen our paper with your suggestions. First, we changed the title of our paper to better reflect the content and the fact that this is a review paper.

In addition, we have added narrative (and references) to strengthen the fact that these conditions can exist independently but also have shared explanatory neurobiological mechanisms that account for the complex clinical problems. 

Round 2

Reviewer 1 Report

This has been greatly improved and I believe now can offer some interesting insights into improving our treatment of the mTBI.

The problems now are mainly minor editing and English language - I began doing this with comment boxes and highlighted text but it is necessary to edit this article thoroughly before it is ready for publication. There are some simple mistakes such as the position of the full stop before the citation and the use of numerals rather than being written in full.

Author Response

Dear Reviewer,

We sincerely appreciate your interest and tremendous dedication and help with critiques, edits, and suggestions for our manuscript. We have amended the manuscript accordingly and have incorporated many of your edits and tried to address as many of your suggestions as possible. We hope that this latest revision will be more readable and satisfactory.

Reviewer 3 Report

The authors have addressed the majority of the reviewer's comments.

Not much data is presented, which is a bit disappointing, however.

Author Response

Dear Reviewer,

We appreciate the opportunity to submit our latest revised manuscript.